

# A Model Instability Issue in the NCEP Global Forecast System Version 16 and Potential Solutions

Xiaqiong Zhou[1, 2] and Hann-Ming Henry Juang[1, 3]

1. NOAA/NCEP/EMC, College Park, MD, USA
2. NOAA/OAR/Geophysical Fluid Dynamics Laboratory, Princeton, NJ, USA
3. Institute of Atmospheric Physics, National Central University, Jongli, Taiwan, ROC

Correspondence to: Xiaqiong Zhou (xiaqiong.zhou@noaa.gov)

**Abstract:** The National Centers for Environmental Prediction (NCEP) Global Forecast System (GFS) version 16 encountered a few model instability failures during the pre-operational real-time
parallel runs. The model forecasts failed when an extremely small thickness depth appeared at the model's lowest layer during the landfall of strong tropical cyclones. A quick solution was to increase the value of minimum thickness depth, an arbitrary parameter introduced to prevent the occurrence of extremely thin model layers, thus numerical instability. This modification solved the issue of the model's numerical instability with a small impact on forecast skills. It was adopted
in GFSv16 to help implement this version of the operational system as planned.

Further investigation showed that the extremely small thickness depth occurred after the advection of geopotential heights at the interfaces of model layers. In the FV3 dynamic core, the horizontal winds at interfaces for advection are calculated from the layer-mean values by solving a tridiagonal system of equations in the entire vertical column based on the Parabolic Spline
Method (PSM) with high-order boundary conditions (BCs). We replaced the high-order BCs with zero-gradient BCs for the interface-wind reconstruction. The impact of the zero-gradient BCs was investigated by performing sensitivity experiments with GFSv16, idealized mountain ridge tests, and the Rapid Refresh Forecast System (RRFS). The results showed that zero-gradient BCs can fundamentally solve the instability and have little impact on the forecast performances and the
numerical solution of idealized mountain tests. This option has been added to FV3 and will be utilized in the GFS (GFSv17/GEFSv13) and RRFS for operations in 2024.

## 1. Introduction

The Next Generation Global Prediction System (NGGPS) of the National Centers for Environmental Prediction (NCEP) is evolving into the Unified Forecast System (UFS). It is
designed to be the source system for NOAA's operational numerical weather prediction



applications and acts as the foundation to better align collaboration with the U.S. modeling community (Ji and Toepfer 2016). The Geophysical Fluid Dynamics Laboratory (GFDL) Finite-Volume Cubed-Sphere (FV3) was chosen as the dynamical core for NGGPS in 2016 (Putman and Lin, 2007; Harris and Lin, 2013). The first major NGGPS model package was successfully implemented within the Global Forecast System (GFS). It became operational on 12 June 2019 as the GFS version 15 (referred to as GFDv15) to replace a legacy spectral model. The GFS was updated from version 15 to 16 (referred to as GFSv16) on 22 March 2021 with an increased number of vertical layers and model physics upgrades.

The retrospective and real-time experiments, covering part of the 2018 hurricane season and the period from May 10, 2019, to real-time before the official implementation was carried out to comprehensively evaluate the forecast performance of GFSv16. GFSv16 showed improved forecast skills compared with GFSv15 in many aspects such as better 500-hPa height anomaly correlation scores and synoptic patterns in the medium range, a better position of relevant frontal boundaries, reduced low-level cold bias during the cool season, and improved Quantitative Precipitation Forecast (QPF) Equitable Threat Scores (ETS) and biases in the medium range.

GFS is the most important operational global weather forecast system at NCEP/EMC. It is not only widely used around the world, but also most of NCEP's forecast systems depend on GFS products. The stability of this operational system is critical to delivering reliable real-time products to its users and downstream forecast systems. GFSv16 encountered model instability issues as several cases during the real-time parallel runs crashed before reaching a 16-day forecast length. The diagnosis of the problematic cases in GFSv16 and corresponding proposed fixes are summarized in this study. The numerical model used in GFSv16 is introduced in Section 2. The diagnostic results are summarized in Section 3. Two potential solutions to fix model instability issues are discussed in Section 4. Section 5 introduces the impact of proposed fixes on forecast performances with sensitivity experiments. Summary and discussion are provided in Section 6.

## 2. Model configuration upgrades

GFSv16 uses a GFDL FV3-based model as its previous version GFSv15. A detailed description of the FV3 dynamic code can be found in the published papers of the GFDL FV3 team (Lin and Rood, 1997; Lin 2004; Harris and Lin 2013; Putman and Lin, 2007; Harris et al. 2020ab). Only a short summary is given here.



The GFDL FV3 uses the equidistant gnomonic projection (Putman and Lin 2007), which splits each cube edge into N equally sized segments and generates a regular mesh on a sphere by connecting non-orthogonal coordinate lines along great circles between two opposite cubic edges.

There are two levels of time-stepping inside FV3. The inner time step (also referred to as the acoustic time step) is the integration of the dynamics along the Lagrangian surfaces, which includes computing the forward in-time horizontal flux terms along the Lagrangian surface, and the pressure-gradient force and elastic terms evaluated backwards-in-time. The outer time step is the vertical remapping process to re-grid the deformed Lagrangian surface to a reference

coordinate.

The governing equations in FV3 in each horizontal layer are fully-compressible flux-form vector-invariant Euler equations (Harris and Lin, 2013). The momentum flux transportation is represented as vorticity flux and the gradient of the kinetic energy without gradients of vectors. The horizontal discretization of FV3 is derived using a two-grid system with the

prognostic winds staggered on a D-grid and C-grid winds used to calculate the face-normal and time-mean fluxes across the cell interfaces (Lin and Rood, 1997). The C-grid winds are interpolated from D-grid winds and then advanced a half time step as the D-grid, except with lower-order fluxes, for efficiency.

The scalar advection scheme is based on the piecewise-parabolic method (PPM; Collella

and Woodward, 1984) with a two-dimensional combination of one-dimensional flux methods (Lin and Rood, 1996). The same subgrid reconstruction unlimited scheme is used for mass, potential temperature, vorticity and momentum. The transport of tracers uses a simplified monotonicity constraint (Lin and Rood, 1997) and Huynh's second-order constraint (Putman and Lin, 2007).

The evaluation of the pressure-gradient force in FV3 remains a 4[th]-order accuracy and is

consistent with Newton's 3[rd] law of motion and achieved by finite-volume integration about a grid cell (Lin 1997). The "Vertically Lagrangian" dynamics of Lin (2004) were extended with the non-hydrostatic pressure gradient computation of Lin (1997) and included a traditional semi-implicit solver for fast vertically propagating sound waves and gravity waves with efficient computation and great accuracy.

The Lagrangian vertical coordinate (Lin, 2004) is one unique aspect of the FV3, in which each vertical layer resembles that of a shallow water system and is allowed to deform freely during the horizontal integration. It is periodically remapped by vertically redistributing mass,





momentum, and energy to a predefined Eulerian coordinate to prevent severe distortion of the Lagrangian surfaces. Vertical transport occurs implicitly from horizontal transport along
Lagrangian surfaces.

GFSv16 is built on 13-km quasi-uniform grids having six tiles globally with each tile having 786×786 grid cells. The physics time step is 150 seconds. In GFSv16, the "remapping" time step is 75 seconds and the shortest acoustic timestep is 12.5 seconds. The model uses the sigma pressure hybrid coordinate with near surface sigma levels, blended sigma/constant pressure
levels in mid-atmosphere, constant pressure levels above.

The major upgrade of GFSv16 from GFSv15 includes an increased number of vertical layers from 64 and 127 with the extended model top from 54 km to 80 km and physics upgrades. The upgraded physics parameterization includes a new scheme to parameterize both stationary and non-stationary gravity waves (Alpert et al. 2019; Yudin et al. 2016, Yudin et al. 2018),  a new
scale-aware turbulent kinetic energy-based moist eddy-diffusivity mass-flux vertical turbulence mixing scheme to better represent the planetary boundary layer processes (Han and Bretherton, 2019), the improved solar radiation absorption by water clouds and the cloud-overlapping algorithm for the Rapid Radiative Transfer Model for GCMs (RRTMG) (Iacono et al., 2008), and improved GFDL cloud microphysics for computing ice cloud effective radius (Harris et al.,
2020ab, Zhou et al. 2019).

### 3.  The study of failed cases

There were eight failed cases during the GFSv16 retrospective and real-time parallel run. They are the cases with the forecast starting times at 00Z 22 Sep. 2018, 18Z 22 July 2020, 06Z 2 Sep.
2020, 06Z and 18Z 3 Sep. 2020, 12Z 4 Sep. 2020, 06Z 5 Sep. 2020, 00Z 6 Sep. 2020 respectively.

A series of sensitivity tests were performed to increase the model stability. Several methods available in FV3 for numerical diffusion to maintain model stability and control energy cascading were tested. For example, a Rayleigh damping method can be used to dampen the winds to zero with the shortest timescale (tau) at the top increasing with pressure until reaching a defined cutoff
pressure level. The minimum timescale and the cutoff pressure level were tuned to apply a stronger Rayleigh damping, but that did not fix the model stability issue for all eight crashed cases. Other parameters such as the non-dimensional divergence damping coefficient, the Smagorinsky-type damping coefficient, and the parameters that control the sponge-layer damping to the top two




layers of the model were also tuned, but these modifications also could not completely solve
instability issues.

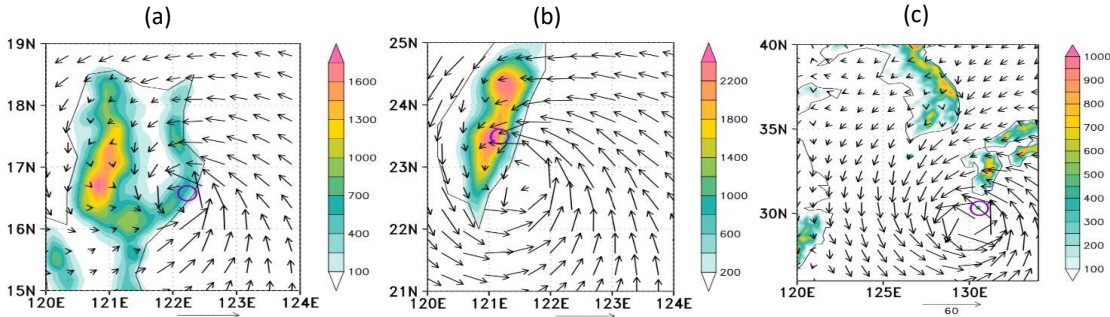

Fig. 1 Wind fields before model crash from the cases with the initial starting time at a) 18Z 22
    July 2020, b) 00Z 22 Sep. 2018 and c) 06Z 2 Sep. 2020. The shading is terrain height
    (unit: m). The open circles mark the location of the crash.

The diagnosis of these cases indicated that all model failures were related to the landfall of
strong tropical cyclones. Negative layer thickness in the pressure between lower and upper
interfaces or not-a-number (NaN) layer thickness in geopotential height was observed. All failures
occurred at grid points located over land when the eyewall of a strong tropical cyclone made
landfall from the east. For example, the forecast starting from 18Z July 22 2020 failed when a
strong tropical cyclone reached the Philippine east coast with strong onshore winds of about 40-
50 m/s (Fig. 1a). In another case, the forecast was interrupted at a grid over the Taiwan Central
Mountain area when a strong tropical cyclone started to make landfall. The other six cases were
related to Tropical Cyclone Haiseng (2020) when it approached Yakushima Island south of Japan
(Fig. 1c).

By examining the model prognostic variables in each acoustic time step (12.5s), we found
that unrealistic downdrafts occurred before the failure of the model integration. Fig. 2 shows that
the vertical motion at the specific grid point increases with time. The updraft greater than 5 m/s
abruptly changes to an unrealistically large downdraft with an amplitude greater than 200 m/s in
one acoustic time step, which directly results in the model failure in the next time step. Fig. 3
shows a similar variation of the vertical motion in the case with the initial forecast time at 0000
UTC on Sep. 22, 2018. Similar phenomena were observed in other six cases related to the landfall
of Haiseng (2020) (not shown).



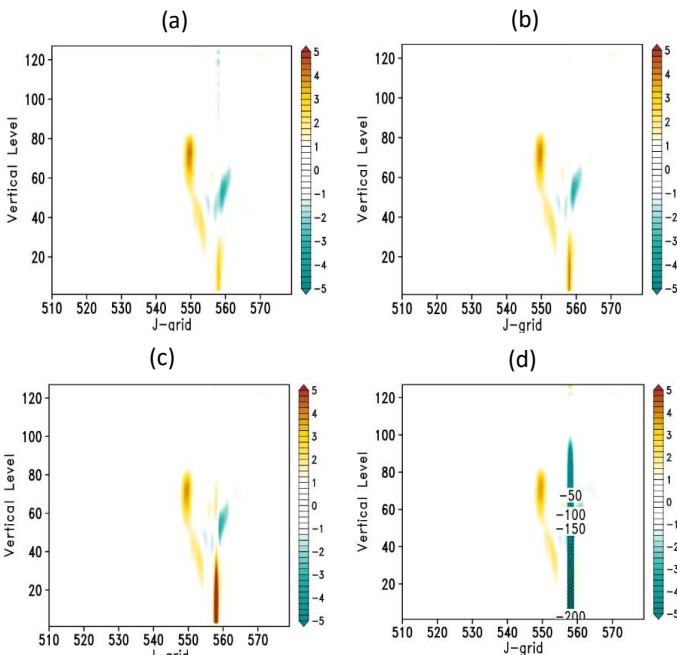

Fig. 2 Vertical section of vertical velocity through the location of the crash along with the model
y-directional grids before the crash for the case with the initial starting time 18Z 18 July
2020. A), b), c), and d) represent 4, 3, 2, and 1 acoustic time step before crash respectively.

The hydrostatic and nonhydrostatic solvers in FV3 are "switchable" at runtime through the namelist option hydrostatic (Harris et al. 2020ab). The nonhydrostatic solver augments the hydrostatic solver by introducing the prognostic variables $w$ and layer height $\delta z$. The pressure thickness $p^*$ is still hydrostatic pressure and nonhydrostatic pressure is diagnosed as a deviation with $p' = p - p^*$ where $p$ is full pressure calculated from the ideal gas law:

$$p = (R_d \theta_v \frac{\partial m}{\partial z})^\gamma \tag{1}$$

Non-hydrostatic pressure perturbation $p'$ and $w$ in the Lagrangian vertical coordinates are solved using a semi-implicit solver, in which the fully implicit time-difference scheme yields a tridiagonal matrix system of equations for $w$. This system requires coefficients and weights related to $p'$ and layer thickness $\delta z$ to solve $w$ with the Thomas algorithm (Thomas, 1949). In the corresponding subroutine for the non-hydrostatic adjustment, the non-hydrostatic pressure perturbation is calculated first where $\theta_v$ is virtual potential temperature. Note that in FV3, all algorithms are formulated in a finite-volume manner: the above variables are cell- or layer-means.



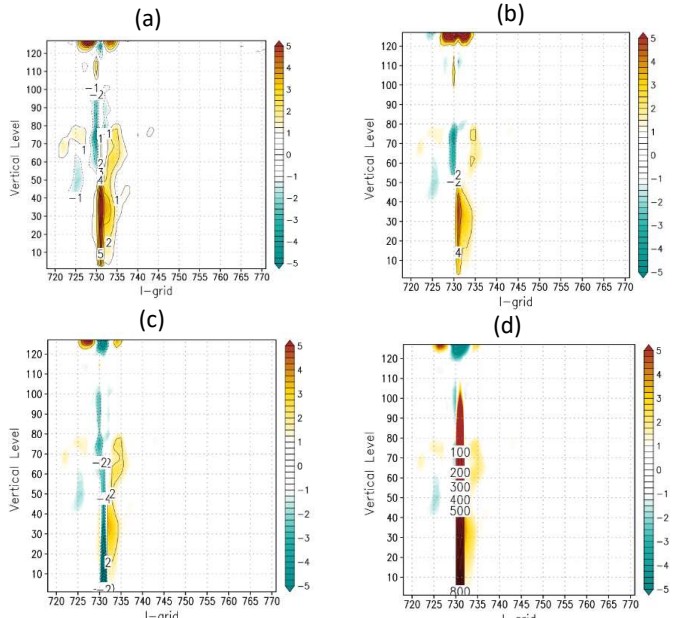

Fig. 3 As Fig. 2 except for the case with an initial starting time of 00Z 22 Sep. 2018 at each
acoustic time step before the crash.


All relevant variables before the crash time including $p^*$, $p$, $\theta_v$, $\delta z$, and the mass $\delta m$ to
calculate $w$ were investigated. Fig. 4 shows that $\theta_v$ and $\delta m$ remain reasonable and consistent before
the crash (Figs. 4c and 4d). The unrealistic value of the full pressure (larger than 5000 hPa) appears
at the model's lowest level at about 200 seconds before the model crash (Fig. 4a), while the

hydrostatic pressure remains reasonable (about 900 hPa) with time (Fig. 4b). The slight
discontinuity of these variables every six acoustic time steps is a result of the vertical remapping
process. GFSv16 has 127 vertical layers with the lowest layer about 20 m thick on average. The
value of $\delta z$ close to 0 200s prior to the crash is quite unusual (Fig. 4e).  The unrealistically
increased $p'$ and the full pressure at the lowest level before the crash come from the occurrence of

extremely small $\delta z$ while computed from the ideal gas law formula. Extremely large downdrafts
are generated through the non-hydrostatic semi-implicit solver from $p'$, which eventually leads to
the model failure. The model instability is a result of the presence of extremely small $\delta z$ at the
lowest model level.



The calculation of $\delta z$, the vertical difference of geopotential height $z$ between the Lagrangian

surfaces before the non-hydrostatic adjustment was investigated. The forward-in-time advective

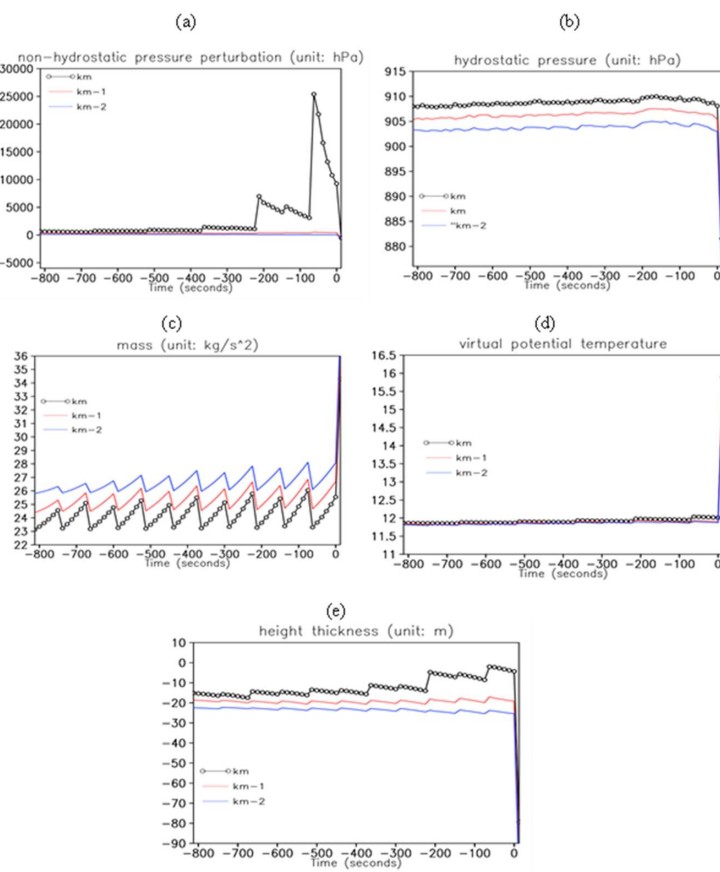

Fig. 4 The time series of a) non-hydrostatic pressure, b) hydrostatic pressure, c) mass, d) virtual
potential temperature and e) thickness depth in height at the crash grid. The black curves
represent the model's lowest level (marked by km), while the blue and red represent the
second and third lowest levels (marked with km-1 and km-2). The open circle marks each
acoustic time step in the time series.

Processes are performed to generate the partially-updated $z$ before the non-hydrostatic adjustment

in the FV3 dynamics. Note that the update of $z$ through advection processes does not directly solve

an equation for the volume of a grid cell ($\delta z$) and it is forward-in-time as the sum of the advective

height flux along with the Lagrangian interfaces and the vertical distortion of the surfaces by the

gradient of $z$. The previous study found that the advection of $\delta z$ created excessive noise near steep



topography (Harris et al. 2021) and it was more difficult to guarantee the kinematic surface condition without perpendicular flow to the surface with the advection of $\delta z$.

To advance $z$ on the interfaces, the advection winds are interpolated from layer means onto the layer interfaces. Fig. 5 shows the time series of $z$ at the crash location before and after the advection process. The value of $z$ at the lowest level before advection remains constant as it is the height of the topography (Fig. 5a). It has a great change after advection and becomes very close to the value of $z$ at the second lowest level (Fig 5b) beginning about 200 seconds before the crash.

There are no significant changes of z before and after the advection at the second and third lowest levels. The advection process at the lowest level is responsible for the decreased thickness depth seen in Fig. 4e.

## 4. Potential solutions

The forward-in-time advection of geopotential height is a part of the acoustic time step in which the Lagrangian surface is allowed to freely deform. An artificial limiter is defined as the minimum thickness depth after the geopotential height advection to enhance its monotonicity in the vertical. This limiter is defined as *dz_min* in FV3 with 2 meters as the default in FV3. It only takes in effect when the thickness of geopotential occurred in the model is smaller than the default

value. We found that increasing this limiter value from 2 to 6 meters can effectively avoid model crashes. All eight cases can finish 16-day forecasts with this modification.

    To examine whether increasing this artificial limiter violates general model states in GFSv16, the possibility for $\delta z$ to reach the minimum thickness depth of 6 was investigated in both crash cases and successful cases. The successful cases were randomly selected from the retrospective

runs among the cases that can complete 16-day forecasts successfully. There were no extremely small $\delta z$ values in any grids from randomly selected successful cases. $\delta z$ less than 6 meters only occurred at the breakpoint in crash cases. This examination indicates that this artificial limiter is only used in very rare situations. Forecast-only experiments also showed that this fix had very little impact on the forecast skill. Since any changes in the forecast performance were not desirable at

the final retrospective test stage for the implementation of GFSv16, this method was considered a suitable temporary fix for GFSv16. It was adopted for the GFSv16 implementation and this fix allowed GFSv16 to be implemented at the time.



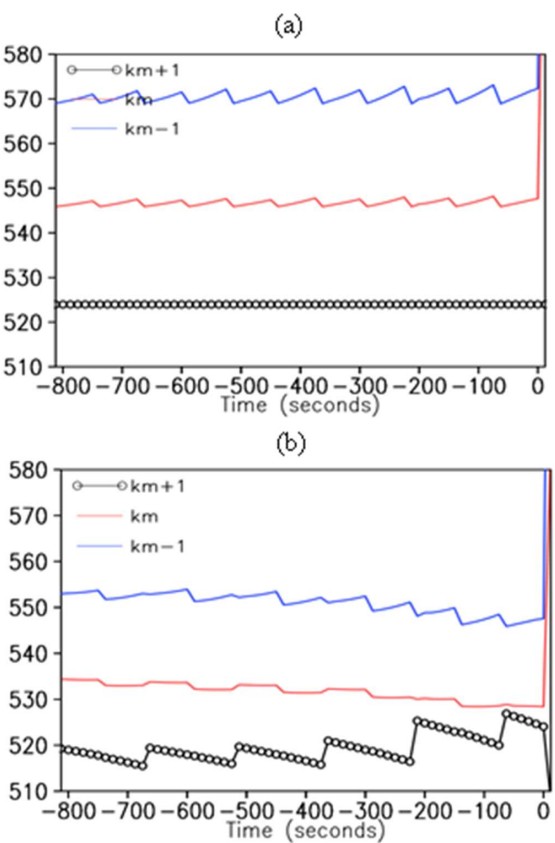

Fig.5 As Fig. 4 except for the geopotential height at the model's three lowest layers (marked as
km+1, km, km-1 respectively) a) before and b) after the advection procedure.

A sensitivity experiment was performed by restarting the model about 1 hour before the crash
with increased minimum thickness depth. The geopotential height after the advection was forced
to be greater than the artificial limiter. The abrupt change of the geopotential height was observed
at the original crash location and time, then it backed to a normal range after several acoustic time
steps (Fig. 6a). The model can successfully finish 16-day forecasts. The increased minimum
thickness depth can prevent the model from crashing, but it does not fundamentally solve the model
instability issue.  In addition, this arbitrary limiter should be used with caution and the height of
the model's lowest level should be considered to select a reasonable value for the limiter.
The advection process to update $z$ in FV3 was examined since the model instability issue likely
originated from the advection of $z$ at the model's lowest level. To update z, the advection winds





and vertical velocity are reconstructed from layer means onto the layer interfaces by solving a tridiagonal system of equations based on the Parabolic Spline Method (PSM, Zerroukat, et al., 2006).

The following equation represents the relationship between the interface value $\hat{q}_{i-\frac{1}{2}}$ and layer-mean value $\bar{q}_i$ ($i$=1, 2, …N) in a computational one-dimensional discretized domain with PSM (Zerroukat et al., 2006):

$$\frac{1}{h_i}\hat{q}_{i-\frac{1}{2}} + 2(\frac{1}{h_i} + \frac{1}{h_{i+1}})\hat{q}_{i+\frac{1}{2}} + \frac{1}{h_{i+1}}\hat{q}_{i+\frac{3}{2}} = 3(\frac{1}{h_i}\bar{q}_i + \frac{1}{h_{i+1}}\bar{q}_{i+1}) \quad (2)$$

where $h_i$ is the spatial interval between two interfaces $h_i = z_{i+\frac{1}{2}} - z_{i-\frac{1}{2}}$ ( i=1, 2...N) and $q$

represents horizontal wind components $u$ and $v$ here. Equations (2) define a linear system of equations for the unknown interface values $\hat{q}_{i-\frac{1}{2}}$ in terms of the layer-mean values $\bar{q}_i$. Boundary

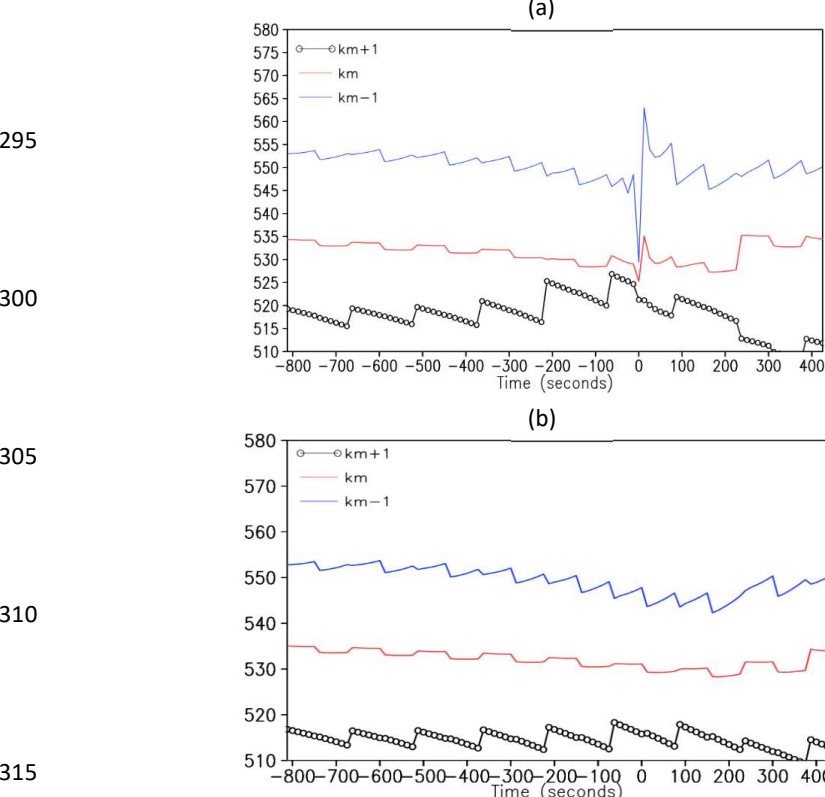

Fig. 6 The time series of the geopotential height at the model's three lowest levels (marked as km+1, km, km-1 respectively) after the advection at the original crash location in the



sensitivity experiments with proposed fixes: a) increased *dz_min* from 2 to 4 and b) zero-gradient BCs.

conditions are required to close the problem. FV3 uses the following equations for the upper and lower boundary to solve the horizontal winds at the model interfaces as a problem of a tridiagonal

system:

$$g_1(g_1 + 0.5)\hat{q}_{\frac{1}{2}} + [1 + g_1(g_1 + 1.5)]\hat{q}_{\frac{3}{2}} = 2g_1(1 + g_1)\bar{q}_1 + \bar{q}_2 \tag{3}$$

$$[1 + g_N(g_N + 1.5)]\hat{q}_{N-\frac{1}{2}} + g_N(g_N + 0.5)\hat{q}_{N+\frac{1}{2}} = \bar{q}_{N-1} + 2g_N(1 + g_N)\bar{q}_N \tag{4}$$

where $g_1 = h_2/h_1$ and $g_N = h_N/h_{N-1}$.

We proposed to use zero-gradient BCs, that is $\frac{dq}{dx}=0$ at the endpoints $z_{\frac{1}{2}}$ and $z_{N+\frac{1}{2}}$ corresponding

to an assumption of zero slope there. Applying these zero-gradient BCs leads to

$$2\hat{q}_{\frac{1}{2}} + \hat{q}_{\frac{3}{2}} = 3\bar{q}_1 \tag{5}$$

$$\hat{q}_{N-\frac{1}{2}} + 2\hat{q}_{N+\frac{1}{2}} = 3\bar{q}_N \tag{6}$$

The original BCs used in FV3 as shown in Eq. (3) and (4) are named as high-order BCs thereafter in contrast with the zero-gradient BCs we proposed.

The comparison of the vertical profiles with two different BCs shows that the reconstructed winds are similar in these two types BCs when the vertical shear of the layer-mean winds in the lower levels is relatively small (Fig. 7a). With larger vertical shear, the overshooting and undershooting of the reconstructed winds at the lowest two layers are more evident by using higher-order BCs than zero-gradient BCs while interior winds remain similar (Fig. 7b). The

vertical shear of interface winds at the lowest several layers are smaller with zero-gradient BCs than with high-order BCs.

With the application of the zero-gradient BCs, all originally crashed cases can finish 16-day forecasts successfully. A sensitivity experiment was performed similarly for zero-gradient BCs. Fig. 6b shows that applying the zero-gradient BCs avoids unrealistic $\delta z$ values. No extremely small

$\delta z$ was found during the model integration. This method is better than increasing the artificial thickness depth limiter as it fundamentally solves the occurrence of unrealistic $\delta z$ values at the model's lowest level.

PSM is third-order accurate in space for a non-uniform grid and fourth-order accurate for a uniform grid (Zerroukat et al. 2006). The reconstructed winds at the BCs with high-order BCs may





retain a relatively higher accuracy. However, it can be worse in the case of sharp/under-resolved gradients with significant overshoots/undershoots due to a larger degree of freedom. Constraints are usually required for the reconstruction to prevent overshoots/undershoots with respect to the layer-mean values (Shchepetkin and McWilliams, 1998; Zerroukat et al. 2006). Our method is reducing the order of the reconstruction polynomial for BCs. It is worth noting that the zero-

gradient condition is only used at the model's upper and lower edge levels. The parabolic spline as the reconstructed function remains valid for the inner layers. In addition, the reconstructed horizontal winds are only used for the advection of geopotential height. The revised BCs do not impact the layer-mean prognostic wind fields directly.

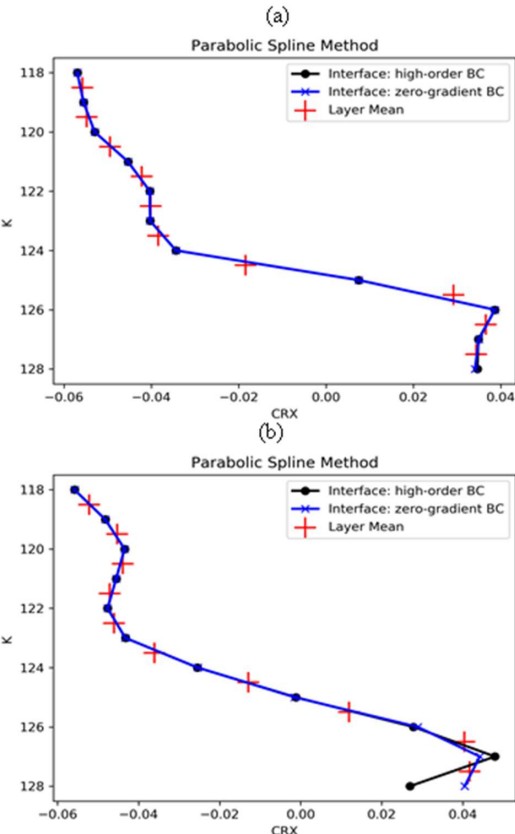

Fig. 7 The vertical profile of Courant numbers in x-axis ($c_x = \triangle t \cdot u / \varDelta x$) at two grids with a) smaller and b) larger vertical gradients. The red cross represents the layer-mean value while the black and blue represent the interface values reconstructed with high-order BCs and zero-gradient BCs.




## 5. Sensitivity experiments with zero-gradient BCs

The impact of zero-gradient BCs on forecast performance was investigated with different model configurations. The experiment design and results are discussed including idealized mountain ridge tests and real-case tests with the same configuration as GFSv16 and the high-

resolution regional application in EMC.

The mountain waves could be sensitive to the model's lower boundary conditions (Smith 2007). The impact of the BC change on the geopotential height advection on the mountain waves was investigated. An idealized mountain ridge test with an adiabatic condition, a uniform flow of 8 m/s over a ridge mountain was performed. This is a modified version of the Dynamical Core

Model Intercomparison Project (DCMIP) case 2.1 with a quasi-2D mountain ridge with a ridge height of 250m (Ullrich et al., 2016). The idealized mountain-ridge experiment was tested on a

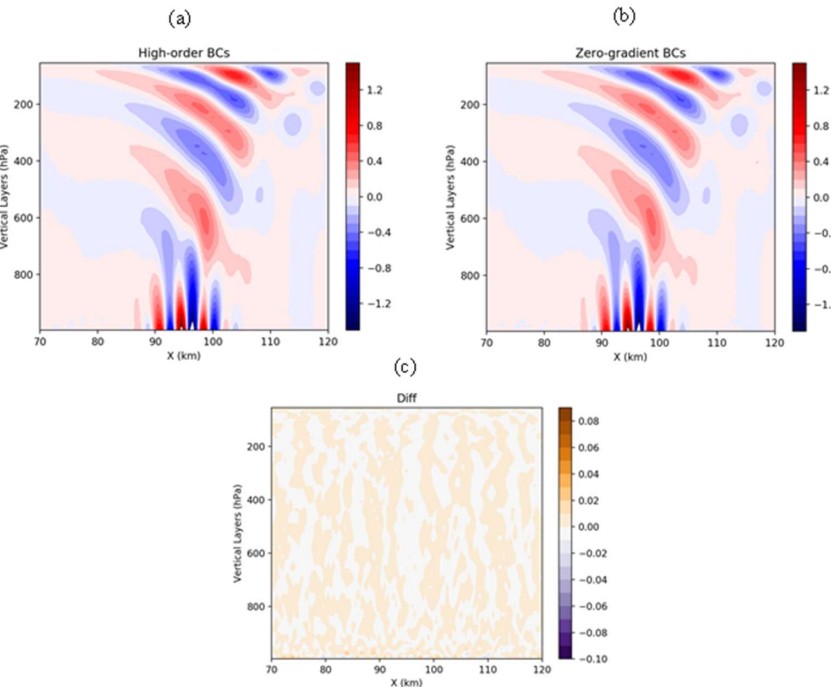

Fig. 8 Cross sections of vertical velocity (m/s) along the equator for orographic mountain ridge
on the earth (quasi-2D ridge in a barotropic zonal flow). The x-axis is longitude (degrees) and the y-axis is the vertical coordinate in pressure (hPa) in (a) the control and (b) the sensitivity run with zero-gradient BCs. c) is the difference between these two runs.



doubly-periodic domain with an adiabatic assumption. The model top is 50 hPa with a horizontal

resolution of 500 m. Zero-gradient BCs are utilized in the upper and lower boundaries in the sensitivity experiment. The mountain wave patterns are similar in these two experiments. The difference between the two zonal wind fields is slightly larger at the lowest levels but remained negligibly small (Fig.8).

A group of sensitivity experiments was performed by using the GFSv16 as the control. A

sensitivity experiment was performed by replacing high-order BCs in the control with zero-gradient BCs at both the lower and upper boundaries of the model to reconstruct horizontal winds with PSM were replaced. 10-day forecasting were compared with initial times from June to October 2020 every five days with 00Z only. The EMC Global NWP Model Verification Package was used for the verification (Yang et al. 2006). This verification package is a standard evaluation

tool for the GFS upgrade and implementation with verification scores comparing gridded model data to both point-based rawinsonde and surface station observations and GFS gridded analysis. The model forecast statistics in terms of the Root Mean Square Error (RMSE), bias, and anomaly correlation for conventional variables, as well as tropical cyclone intensity and track forecasts over the Atlantic, Eastern Pacific, and West Northern Pacific and precipitation threat skill scores over

CONUS. The comparison of these forecast verification metrics shows that the sensitivity experiments with zero-gradient BCs have similar forecast performance without significant differences (not shown) to those of GFSv16 with high-order BCs.

The Rapid Refresh Forecast System (RRFS) is another important FV3-based UFS application in EMC. It is the NOAA next-generation convection-allowing, rapidly-updated ensemble

prediction system, currently scheduled for operational implementation in late 2023. The operational configuration features a 3 km grid spacing covering North America and include forecasts every hour out to 18 hours, with extensions to 60 hours four times per day at 00Z, 06Z, 12Z, and 18Z. Each forecast is planned to be composed of 9-10 members.

The impact of the zerio-gradient BCs on the high-resolution forecasts was also investigated based

on the RRFS configuration. The ensemble members with the Mellor–Yamada–Nakanishi–Niino (MYNN) (Nakanishi 2001; Olson et al., 2019) Planetary Boundary Lateral (PBL) and Thompson MP scheme were used as the control to investigate the impact of zero-gradient BCs. Fig. 9 shows that the precipitation distribution from 12-36 hours in the experiment with zero-gradient BCs resembles that in



the control. The use of zero-gradient BCs does not significantly change the forecast results for high-
resolution forecasts.

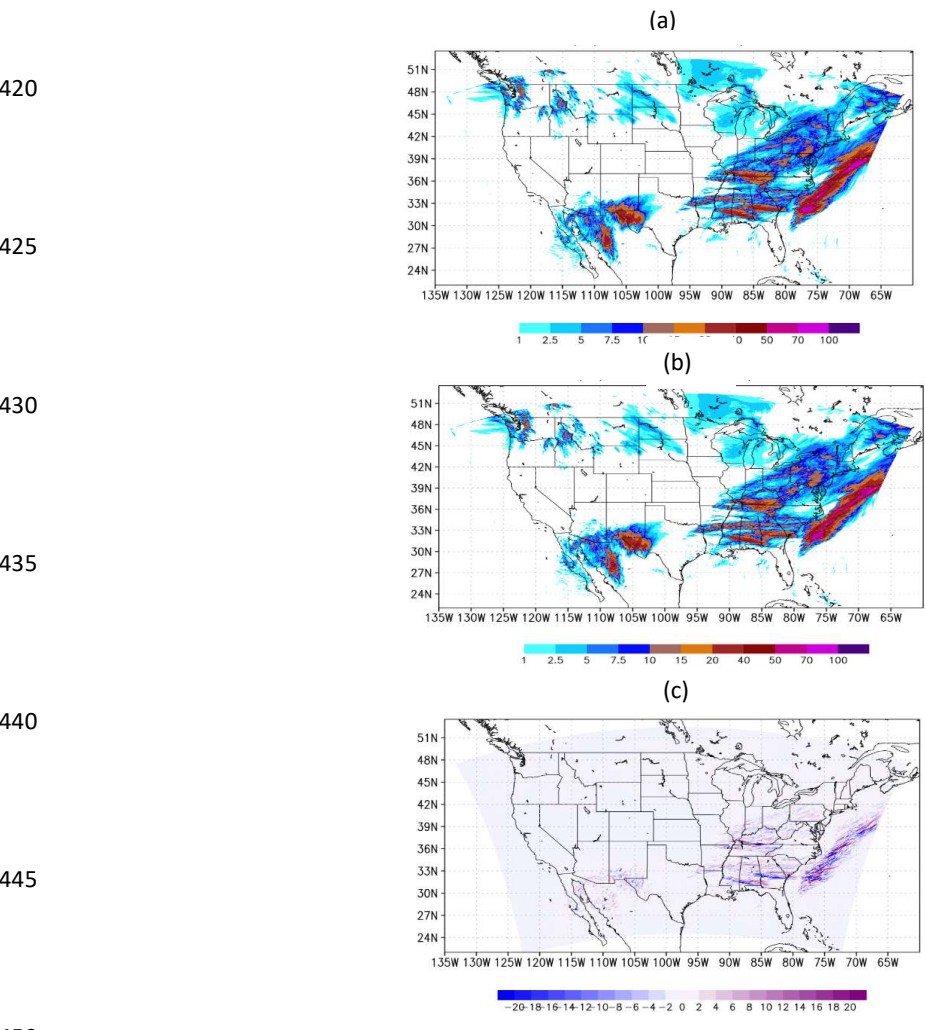

Fig. 9 24-hr precipitation (mm) from 12-36 hr in (a) the control and (b) the sensitivity run with
zero-gradient BCs. c) is the precipitation difference between these two runs. The control run
has the same configuration as the ensemble member one in RRFS with the initial starting
time on 00Z March 2, 2020.



### 6. Summary and discussion

GFS is one of the most important operational global weather forecast systems at NCEP/EMC. The stability of GFS on model integration is as important as its forecast skills to deliver dependable real-time products to its users and downstream forecast systems. The model instability issue of GFSv16 caught our attention when several cases in its real-time parallel runs failed to finish 16-day forecasts. The analysis of these cases showed that the model integration was interrupted after the presence of a very thin layer depth corresponding to a largely deformed layer surface at the model's lowest level in tropical cyclones during the landfall after the advection of geopotential height.

An artificial limiter is defined in FV3 to ensure that the minimum layer depth in FV3 after the advection is not less than a default value to maintain the monotonicity of geopotential height in the vertical. Sensitivity tests showed that increasing the value of this artificial parameter from the default value of 2 to 6 can fix the model instability issue. An abrupt change of geopotential height at the model's lowest interface was still observed with an increased value of the limiter, but all previously crashed cases can finish 16-day forecasts. This method was effective to solve the model instability issue and was adopted to GFSv16 so that the GFSv16 can be implemented as scheduled. Nevertheless, this method lacks a scientific foundation and the root reason corresponding to the model instability remains unknown.

Further investigation suggested that the presence of an extremely thin layer at the model's lowest layer was related to the reconstruction of interface winds from layer mean winds for the advection of geopotential height along the Lagrangian surfaces. In FV3, the horizontal winds are calculated from layer means onto the layer interfaces by solving a tridiagonal system of equations based on PSM (Zerroukat, et al., 2006) with high-order BCs. It was found that the high-order BCs easily produce overshoots or undershoots in areas with large vertical wind shear. The lower boundary in a landfall tropical cyclone was a perfect condition for the occurrence of overshoots/undershoots with high-order BCs, which led to a heavily distorted Lagrangian surface and triggered unstable conditions. The change of BCs from high-order to zero-gradients at the lowest interface removed spurious under/overshoots near steep terrain with vertical wind shears, thus avoiding a distorted geopotential height interface so that the model remains in stable conditions.

The impact of the zero-gradient BCs for the tridiagonal system on the forecast results was very minor. The zero-gradient condition for BCs was only valid at the model's lowest/highest interface. The reconstructed horizontal wind profile at sub-grids remained a parabolic spline as defined in terms of the layer-mean values. In addition, the reconstructed interface horizontal winds were only





used in the advection of geopotential height. The zero-gradient BCs did not impact the prognostic layer-mean wind fields directly. The zero-gradient BCs had been committed to the Unified Forecast System (UFS) as an alternative method for the forward-in-time advection of geopotential height.

Even though the model instability issue only was found during the landfall of tropical storms 495 in GFSv16, it could be the case in any situations with strong vertical shear of winds at the lower and upper boundary. It was found that the zero-gradient BCs can effectively improve the model forecast stability for RRFS in non-tropical cyclone cases. This option has been included in the RRFS package for the real-time parallel runs. For the GFS, the artificial limiter used in GFSv16 will be replaced by the option of zero-gradient BCs to stabilize the model forecasts in the next- 500 generation coupled GFSv17/Global Ensemble Forecast System (GEFS v13). Both the RRFS and GFSv17/GEFSv13 target operational implementation in 2024.

**Code/Data availability**: The numerical model simulations upon which this study is based are too 505 large to archive or to transfer. Instead, we provide all the information needed to replicate the simulations; we used the model version GFSv16. The model code, compilation script, the scripts to run the model and the namelist setting are available at  NOAA-EMC/global-workflow at gfs.v16.2.2 (github.com). The initial condition files used in this study are the GFS/GDAS analysis data but only the recent production is available for the public at  Index of /data/nccf/com/gfs/prod 510 (noaa.gov). Two potential fixes we discussed in this paper can be tested by adding *dz_min* or *psm_bc* in the model input namelist. The model source code, workflow and the scripts and data to plot the figures in this manuscript are also available on Zenodo (https://doi.org/10.5281/zenodo.7555839).

**Author contribution:** Xiaqiong Zhou and Hann-Ming Henry Juan designed the experiments and 515 Xiaqiong Zhou carried them out. Xiaqiong Zhou prepared the manuscript with contributions from the co-author.

**Acknowledgment:** We thank the GFDL modeling group for their support, especially Lucas Harris, 520 Xi Chen and Linjiong Zhou, and EMC colleagues as well, such as Fanglin Yang, Sajal Kar, and



others for their insightful suggestions and discussions. We also thank Miodrag Rancic and Kevin
Viner for their careful EMC internal review.



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
