# Peer review of "A Model Instability Issue in the NCEP Global Forecast System Version 16 and Potential Solutions"

_EGUsphere, 2022_

## Referee Comment (RC1)

**Review of** "A Model Instability Issue in the NCEP Global Forecast System Version 16 and Potential Solutions", by X. Zhou and H.-M. H. Juang.

egusphere 2022-1235

Recommendation: publish after minor revisions

**General evaluation**

The authors discuss a numerical stability issue in the FV3 dycore that appeared several times during the preoperational phase of NCEP's GFS v16, where the transition from the former spectral hydrostatic dycore to the nonhydrostatic FV3 was prepared. The instability was related to extremely small thicknesses of the lowest model layer and could be solved by changing the lower boundary condition for geopotential advection from high-order to zero-gradient. Since the potential occurrence of this type of instability is specific to the numerical implementation of FV3, its solution has probably no relevance for other atmospheric models, but one may argue that the FV3 is sufficiently widely used in the meteorological community to warrant publication in a peer-reviewed journal despite this caveat. I will leave the decision upon the question of sufficient general relevance to the editor. Apart from that, I have only a couple of minor comments.

**Specific comments**

1. p. 3, ln. 84/85: Is the pressure-gradient discretization in FV3 fourth-order accurate including the metric terms related to the coordinate transformation?

2. p. 4, ln. 121: "but that did not fix the model stability issue for all eight crashed cases": That means, it did help for some of the cases, or for none of them? Please clarify.

3. p. 5, ln. 124/125: As the instability was related to a collapse of the surface layer thickness, did the changes of the Rayleigh damping / sponge layer beneath the model top have any impact at all? If yes, how can this be explained?

4. p. 5, caption of Fig. 1: From which model level are the wind fields taken? And the unit for the reference arrow is missing (hopefully m/s).

5. p. 6, Eq. (1): What is $m$? I suppose mass, but this should be indicated here.

6. p. 7, ln. 218: "close to 0 200s ..." please write "zero".

7. p. 8, Fig. 4 and corresponding caption: please indicate the unit of virtual potential temperature and explain the meaning of its values. Is it a deviation from a reference state? In addition, it appears quite odd that the height thicknesses in panel (e) are generally negative. This may be a model-internal sign convention, but a thickness is usually a positive-definite quantity, and it would be better to adjust the figure accordingly.

8. p. 8, ln. 234–237: The sentence starting with "Note that the update" is hard to understand and should be formulated more clearly.

9. p. 12, ln. 329: $\frac{dq}{dx} = 0$ – analogous in $y$-direction? Moreover, I would expect that zero slope means $\frac{dh}{dx} = 0$. Please clarify.

10. p.15, ln. 391: Is replacing upper and lower boundary conditions at the same time needed for consistency, or would it be sufficient to use zero-gradient for the lower bc's?

11. p.15, ln. 392: "forecasting" $\rightarrow$ "forecasts"

12. p. 15, ln. 409: "zerio" → "zero"

13. p. 15, ln. 411: "Planetary Boundary Lateral" → "... Layer"

14. p. 17, ln. 469: "value of 2 to 6 can ..." insert "meters"

---

## Referee Comment (RC2)

**Review of the GMD / egusphere-2022-1235 manuscript:**
A Model Instability ssue in the NCEP Global Forecast System Version 16 and Potential Solutions

**Authors:** Xiaqiong Zhou and Hann-Ming Henry Juang

**Summary:**
The manuscript describes that a numerical detail in the nonhydrostatic cubed-sphere finite-volume dynamical core FV3 triggered numerical instabilities in the NOAA/NCEP operational weather forecast model GFS version 16. These instabilities occurred for rather extreme atmospheric conditions, such as landfalling tropical cyclones that approach mountain ranges. The authors describe the source of the problem which is connected to FV3's floating Lagrangian vertical coordinate. The numerical approach utilizes pressure and height thicknesses between the surrounding interface levels in each model layer which get periodically remapped to a reference coordinate. However, when instabilities occurred, it was diagnosed that the thicknesses had turned negative before the remap step. Two solutions are proposed. One solution is to remap more frequently by enforcing a stricter lower bound for the height thickness. The second approach is a modification of the boundary condition for an implicit tridiagonal solver. The latter is the more general solution and will be included in the forthcoming version GFS version 17. The manuscript is easy to read and informative, especially for the dynamical core research community. The research topic is narrow but should trigger enough interest to warrant a publication. The research is of high quality. However, the write-up in the manuscript contains many sloppy inaccuracies and inconsistencies (e.g., switching sign conventions, missing or confusing axis labels, legends, symbols, or physical units) that need to be addressed before a publication can be recommended. More rigorous proof-reading could have avoided most mistakes. In addition, the quality of some (fuzzy) figures is poor. These issues are detailed below.

**Details:**
1) Lines 62-64: avoid 1-sentence paragraphs
2) Line 97: quoted number of grid points is incorrect, should read 768 x 768
3) Lines 116-125: This section contains a discussion about sponge-layer diffusion mechanisms near the model top and triggers the impression that the crashes are caused by processes near the model top. However, later only the three lowest model levels are analyzed. Clarify where is crashes are triggered in this section. In addition, clarify why the discussion of the sponge-layer diffusion settings are relevant for the analysis in this paper. If they have no relevance, this needs to be clearly stated.
4) Line 124: this point is related to the previous point. What does 'not completely' mean here? It reads as if there were examples when the modification of the sponge-layer mechanisms were able to prevent the crash, but this was not a reliable modification. Clarify this.
5) Line 140 and many others: A sentence cannot start with the abbreviation 'Fig. XX …', use the word 'Figure' at the beginning of sentences.
6) Fig. 2, Fig. 3, Fig. 7, and definitions in the text (line 286): clarify whether level 1 is at the surface or the model top, e.g. are the vertical levels counted downwards or upwards? Figures 2 and 3 suggest that level 1 is at the surface, but Fig. 7 reverses the order of the levels and shows level 127 at the bottom (surface?). This also has implications for the computation of

the layer thickness δz, as e.g. defined in line 289. If layer 1 is the lowest as indicated by Figs. 2 and 3, then the definition of the height thickness becomes negative as also shown in Fig.4e. However, throughout the manuscript the impression is triggered that the height thickness is a positive quantity since its thresholds dz_min (line 253, 255) are always quoted as positive thresholds. This is contradicted by the positions of the lowest and second/third lowest model levels in (also in Figs. 5 and 6) that show that the geopotential height of the lowest level is indeed near the surface (providing negative thicknesses according to line 299). There is general confusion about the sign of δz which needs to be remedied. The line-up of the vertical levels needs to be unified in Figs. 2/3 and 7.

7) The confusion about the level count for the vertical levels also has implications for the legends in Figs. 4, 5, and 6. If level 1 is the lowest level, it is very confusing to name the second lowest level km-1 in the Fig. 4 legend. It should read km+1 in this case. There is general confusion how the legends of these three figures refer to the levels. In Fig. 4, the labels 'km', 'km-1', 'km-2' are used for the lowest, second lowest and third lowest levels, respectively. Note that there is a typo in Fig. 4b, that lists 'km' twice and leaves out km-1. However, in Figs. 5 and 6 the labels change to 'km+1' for the lowest level (was km before), 'km' for the second lowest (was km-1) and 'km-1' for the third lowest (was km-2). If 'km' denotes the total number of levels like 127 in Fig. 4, then level 127 is located at the surface in contradiction to Figs. 2 and 3 and level numbers decrease upwards. This all needs to be made consistent with unified legends and a unique way how levels are counted.

8) Figures 2 and 3 contain confusing x-axis labels. What are the label J-grid and I-grid? Switch to some readable labels like 'latitudes' or 'longitudes'. Do these figures show cross sections along interpolated latitudes or longitudes, or data along cubed-sphere coordinates? The captions of Fig. 2 and 3 need to list the physical unit of w. clarify whether this is the vertical height velocity or vertical pressure velocity. I assume it is the height velocity, but die to the missing unit this is not clear.

9) Line 175-176: inaccurate definition of the symbols δz and p*. δz is defined as the layer height, but this is the symbol for the layer thickness. Explain how the layer thickness is computed and whether it is negative or positive. Line 175 needs to state the physical meaning of the symbol 'w' for clarity. In addition, the symbol p* is incorrectly defined as the hydrostatic layer thickness, but p* symbolizes the hydrostatic pressure (not the thickness). The symbol δp* needs to be used for the hydrostatic pressure thickness.

10) Line 178: all symbols in Eq. (1) need to be explained including $R_d$, m, z and γ. Is $R_d$ the moist or dry gas constant

11) Line 211 needs to explain the definition of the symbol δm. Again, is this considered a positive or negative quantity in connection with δz? In Fig. 4c, δm is positive, but δz is negative in Fig. 4e which renders a quantity like δm/δz negative. A negative quantity like this will not work for the computation of the pressure in Eq. (1), leading to imaginary parts. If the authors insist on these sign conventions, Eq. (1) seems to be wrong.

12) Line 218: should read '0-200'

13) Figs, 4, 5, 6 captions: specify which test case this is and refer to the location of the circle in Fig. 1 (which subfigure?)

14) Fig. 4b: it is unusual to plot pressure from lower pressure to higher pressure along the y-axis. The higher pressure is at the lower location, so the pressure axis needs to be reversed starting from the higher pressure decreasing along the y-axis.

15) Fig. 4c, the title of the plot 'mass' and the caption are incorrect. This is $\delta m$ and not m. Correct. The physical unit $kg/s^2$ does not make sense. Do you mean $kg/m^2$ ?

16) Fig. 4d: The values for the virtual potential temperature in the range 11-16.5 (which physical unit? units are omitted, add the units) do not make physical sense. Potential temperatures near the surface lie around 300 K depending on location. It is likely that there is another (sloppy) oversight when defining the symbol $\theta_v$. It is not the actual virtual potential temperature, but a scaled version of it. The definition needs to be shown when referring to it in line 184.

17) Fig 4: all legends are too small, enlarge the font size and line thickness

18) Figs. 5 and 6: specify the physical unit along the y-axis.

19) Lines 323: here, the layer numbering suggests that level 1 is at the top of the atmosphere since its boundary condition is described as the 'upper' boundary condition. As specified earlier, this all needs to be remedied.

20) Fig. 7: Does 'K' denote the level number? This is undefined. The ordering switches in comparsion to Figs. 2 and 3 with the level number 127 at the bottom (surface?). This needs to be cleaned up (see also the earlier comments).

21) Line 374: what is the modification? Specify it to make the test reproducible.

22) Fig. 8c: the color range looks inadequate, but maybe there are tiny (invisible) points that range over this scale. Clarify. The caption states that the x-axis has units of degrees, but the actual axis label states 'km'. Correct this inconsistency. Add the physical units.

23) Line 386: does this implementation utilize a sponge layer near the model top? If yes, descrive it.

24) Line 400: An analysis is discussed, but no figure or scores are provided. Is this intentional or an oversight? If no figure was planned state '(not shown)'.

25) Line 409 typo 'zerio'

26) Lines 507 and 509: The phrases 'NOAA-EMC/global-workflow at gfs.v16.2.2 (github.com)' and 'Index of /data/nccf/com/gfs/prod (noaa.gov)' look like links, but they are not functional. Correct this.

27) Line 541: Point to the newer version of the Harris et al FV3 model description (Technical document from June 2021). The provided link leads to an empty page. It is incorrect to also state that this was published in J. Adv. Model Earth Sys.,12 (10), 2020b (remove the journal name)

28) Lines 525 & 584: incomplete references, both needs the location and conference dates. Capitalize fv3gfs and fv3.

29) Figs. 4, 5, 6, 7 are rather fuzzy and of low quality. Improve the quality.

---

## Author Comment (AC1)

Thank you very much for your time and your comments and suggestions. We appreciate your help to improve our manuscript. The following is our point-to-point replies:

1.p. 3, ln. 84/85: Is the pressure-gradient discretization in FV3 fourth-order accurate including the metric terms related to the coordinate transformation?
Yes, the pressure-gradient discretization in FV3 with fourth-order accuracy includes the metric terms related to the coordinate transformation. The discretization in FV3 uses a finite-volume integration method to compute the pressure gradient force in general vertical coordinates (Lin 1999). It is based on fundamental physical principles in the discrete physical space, rather than on the common approach of transforming analytically the pressure gradient terms in differential form from the vertical physical (i.e., height or pressure) coordinate to one following the bottom topography. This is analogous to the finite-volume of the advective process in which the accuracy depends on the assumed subgrid distribution of the advected constituent.

2. p. 4, ln. 121: "but that did not fix the model stability issue for all eight crashed cases": That means, it did help for some of the cases, or for none of them? Please clarify.
It helped for some of the cases. The line 121 is revised to emphasize that some cases can finish 16-day.

3. p. 5, ln. 124/125: As the instability was related to a collapse of the surface layer thickness, did the changes of the Rayleigh damping / sponge layer beneath the model top have any impact at all? If yes, how can this be explained?
It is a good question. You are right that the changes of Rayleigh damping/sponge layer near the model top cannot solve the issue related to the collapse of the surface layer thickness. We believed that the occurrence of surface layer thickness is a rare and extreme event in the model. Applying stronger damping near the model top likely change the forecasts with model integration and we are just "lucky" to avoid the crash.

4. p. 5, caption of Fig. 1: From which model level are the wind fields taken? And the unit for the reference arrow is missing (hopefully m/s).
The wind fields are at the model lowest layer with unit (m/s). It is added in Fig. 1 caption.

5. p. 6, Eq. (1): What is m? I suppose mass, but this should be indicated here.
Yes, m is mass, it is added in the manuscript

6. p. 7, ln. 218: "close to 0 200s ..." please write "zero".
Revised.

7. p. 8, Fig. 4 and corresponding caption: please indicate the unit of virtual potential temperature and explain the meaning of its values. Is it a deviation from a reference state? In addition, it appears quite odd that the height thicknesses in panel (e) are generally negative. This may be a model-internal sign convention, but a thickness is usually a positive-definite quantity, and it would be better to adjust the figure accordingly.
The definition of the virtual potential temperature in FV3 is added and its unit is indicated in figure 4. height thinkness is revised to positive value in the figure 4(e).

8. p. 8, ln. 234–237: The sentence starting with "Note that the update" is hard to understand and should be formulated more clearly.
This sentence was rewritten

9. p. 12, ln. 329: $\frac{dq}{dx}$ 0 – analogous in y-direction? Moreover, I would expect that zero slope means
$\frac{dh}{dx}$ = 0. Please clarify.

$\frac{dq}{dx}$ should be $\frac{dq}{dz}$. It is corrected.

10. p.15, ln. 391: Is replacing upper and lower boundary conditions at the same time needed for consistency,  or would it be sufficient to use zero-gradient for the lower bc's?

These two kinds of BCs make have little different impact on the model upper BCs. But we replaced both lower and upper boundary for consistency.

11. p.15, ln. 392: "forecasting"!"forecasts"

Revised to forecasts

12. p. 15, ln. 409: "zerio"!"zero"

Corrected

13. p. 15, ln. 411: "Planetary Boundary Lateral"!"... Layer"

Corrected

14. p. 17, ln. 469: "value of 2 to 6 can ..." insert "meters"

"meters" are added.

---

## Author Comment (AC2)

Reply:

Thank you for the excellent summary of our work. Your summary accurately and concisely captured the key aspects and conclusions we would like to address in our manuscript. We revised the manuscript carefully according to your comments and suggestions. The following is our point-to-point replies:

**Details:**
1) Lines 62-64: avoid 1-sentence paragraphs
   The paragraphs was revised, more details were added.
2) Line 97: quoted number of grid points is incorrect, should read 768 x 768
   Corrected.

3) Lines 116-125: This section contains a discussion about sponge-layer diffusion mechanisms near the model top and triggers the impression that the crashes are caused by processes near the model top. However, later only the three lowest model levels are analyzed. Clarify where is crashes are triggered in this section. In addition, clarify why the discussion of the spongelayer diffusion settings are relevant for the analysis in this paper. If they have no relevance, this needs to be clearly stated.

This paragraph was revised. We tuned many tunable available parameters to stabilize the model. Although some of the cases were able to finish 16-day forecasts, not all of them became stable with these tuned parameters, indicating that further improvements were needed to address the model stability issue. That's why we looked into the model results and found the problem is actually located near the model lower boundary.

4) Line 124: this point is related to the previous point. What does 'not completely' mean here? It reads as if there were examples when the modification of the sponge-layer mechanisms were able to prevent the crash, but this was not a reliable modification. Clarify this.

We clarified that in the manuscript. Although some of the cases were able to finish 16-day forecasts, not all of them became stable with these tunings. It indicates that further improvements are needed to address the model stability issue.

5) Line 140 and many others: A sentence cannot start with the abbreviation 'Fig. XX …', use the word 'Figure' at the beginning of sentences.

Corrected

6) Fig. 2, Fig. 3, Fig. 7, and definitions in the text (line 286): clarify whether level 1 is at the surface or the model top, e.g. are the vertical levels counted downwards or upwards? Figures 2 and 3 suggest that level 1 is at the surface, but Fig. 7 reverses the order of the levels and shows level 127 at the bottom (surface?). This also has implications for the computation of the layer thickness $dz$, as e.g. defined in line 289. If layer 1 is the lowest as indicated by Figs. 2 and 3, then the definition of the height thickness becomes negative as also shown in Fig.4e. However, throughout the manuscript the impression is triggered that the height thickness is a positive quantity since its thresholds $dz\_min$ (line 253, 255) are always quoted as positive thresholds. This is contradicted by the positions of the lowest and second/third lowest model levels in (also in Figs. 5 and 6) that show that the geopotential height of the lowest level is indeed near the surface (providing negative thicknesses according to line 299). There is general confusion about the sign of $dz$ which needs to be remedied. The line-up of the vertical levels needs to be unified in Figs. 2/3 and 7.

Figure 2 and 3 are revised. The level 1 is the model top as in the model. Figs. 2 & 3 are revised to be consistent with Fig. 7. Dz in fig 4e is changed to be positive in order to be consistent with the manuscript "height thickness", dz_min et al.

7) The confusion about the level count for the vertical levels also has implications for the legends in Figs. 4, 5, and 6. If level 1 is the lowest level, it is very confusing to name the second lowest level km-1 in the Fig. 4 legend. It should read km+1 in this case. There is general confusion how the legends of these three figures refer to the levels. In Fig. 4, the labels 'km', 'km-1', 'km-2' are used for the lowest, second lowest and third lowest levels, respectively. Note that there is a typo in Fig. 4b, that lists 'km' twice and leaves out km-1. However, in Figs. 5 and 6 the labels change to 'km+1' for the lowest level (was km before), 'km' for the second lowest (was km-1) and 'km-1' for the third lowest (was km-2). If 'km' denotes the total number of levels like 127 in Fig. 4, then level 127 is located at the surface in contradiction to Figs. 2 and 3 and level numbers decrease upwards. This all needs to be made consistent with unified legends and a unique way how levels are counted.

Fig4b label was corrected. The lowest level is defined with N, the second and third lowestest level are define by N-1, N-2 in the labels. If there are total N layers in the model, there are N+1 levels interface. The captions for Fig5 and 6 are revised to clarify. "km " in Figs.5 and 6 are revised to "N" too.

8) Figures 2 and 3 contain confusing x-axis labels. What are the label J-grid and I-grid? Switch to some readable labels like 'latitudes' or 'longitudes'. Do these figures show cross sections along interpolated latitudes or longitudes, or data along cubed-sphere coordinates? The captions of Fig. 2 and 3 need to list the physical unit of w. clarify whether this is the vertical height velocity or vertical pressure velocity. I assume it is the height velocity, but die to the missing unit this is not clear.

The x-axis labels are revised to X and Y with the reference distance with kilometer (KM) from the crash grid. The data is along cubed-sphere coordinate. The vertical velocity in Figs 2 and 3 is m/s, which is added in Figs. 2 and 3 captions.

9) Line 175-176: inaccurate definition of the symbols dz and p*. dz is defined as the layer height, but this is the symbol for the layer thickness. Explain how the layer thickness is computed and whether it is negative or positive. Line 175 needs to state the physical meaning of the symbol 'w' for clarity. In addition, the symbol p* is incorrectly defined as the hydrostatic layer thickness, but p* symbolizes the hydrostatic pressure (not the thickness). The symbol dp* needs to be used for the hydrostatic pressure thickness.

The definition of dz and p* are corrected. dz is layer thickness and p* is hydrostatic pressure. W is vertical velocity.

10) Line 178: all symbols in Eq. (1) need to be explained including $R_d$, m, z and g. Is $R_d$ the moist or dry gas constant

The definition of these variables are added.

11) Line 211 needs to explain the definition of the symbol dm. Again, is this considered a positive or negative quantity in connection with dz? In Fig. 4c, dm is positive, but dz is negative in Fig. 4e which renders a quantity like dm/dz negative. A negative quantity like this will not work for the computation of the pressure in Eq. (1), leading to imaginary parts. If the authors insist on these sign conventions, Eq. (1) seems to be wrong.

Dz in Fig4 is changed to positive value.
12) Line 218: should read '0-200'

This is rewritten to avoid confusing. "0" changed to zero
13) Figs, 4, 5, 6 captions: specify which test case this is and refer to the location of the circle in Fig. 1 (which subfigure?)
Figs 4,5,6 are for the case corresponding to Fig.1a. The initial time of this case is specified in figure caption.

14) Fig. 4b: it is unusual to plot pressure from lower pressure to higher pressure along the y-axis. The higher pressure is at the lower location, so the pressure axis needs to be reversed starting from the higher pressure decreasing along the y-axis.
Y-axis in Fig.4a is flipped following your suggestion.
15) Fig. 4c, the title of the plot 'mass' and the caption are incorrect. This is dm and not m. Correct. The physical unit $kg/s_2$ does not make sense. Do you mean $kg/m_2$ ?
The mass unit is corrected. It should be $kg/m^2$.
16) Fig. 4d: The values for the virtual potential temperature in the range 11-16.5 (which physical unit? units are omitted, add the units) do not make physical sense. Potential temperatures near the surface lie around 300 K depending on location. It is likely that there is another (sloppy) oversight when defining the symbol $q_v$. It is not the actual virtual potential temperature, but a scaled version of it. The definition needs to be shown when referring to it in line 184.
The definition of the virtual potential temperature used in FV3 is added after line 190. FV3 use a reference pressure 1 pa.
17) Fig 4: all legends are too small, enlarge the font size and line thickness
Fig. 4 is replotted to have larger font and clear lines.
18) Figs. 5 and 6: specify the physical unit along the y-axis.
Title for fig 5 and 6 is added in the figure.
19) Lines 323: here, the layer numbering suggests that level 1 is at the top of the atmosphere since its boundary condition is described as the 'upper' boundary condition. As specified earlier, this all needs to be remedied.
Revised.
20) Fig. 7: Does 'K' denote the level number? This is undefined. The ordering switches in comparsion to Figs. 2 and 3 with the level number 127 at the bottom (surface?). This needs to be cleaned up (see also the earlier comments).
The y-axis label "K" is changed to "vertical level". The model level from top to surface has been clarified.
21) Line 374: what is the modification? Specify it to make the test reproducible.
Instead of assuming a small Earth, the idealized mountain-ridge experiment was tested on a doubly-periodic domain. Specific details of the experiments are introduced in the manuscript.
22) Fig. 8c: the color range looks inadequate, but maybe there are tiny (invisible) points that range over this scale. Clarify. The caption states that the x-axis has units of degrees, but the actual axis label states 'km'. Correct this inconsistency. Add the physical units.
X-axis is in km, the caption is corrected. The color scale in Fig8c is revised.
23) Line 386: does this implementation utilize a sponge layer near the model top? If yes, descrive it.

Two forms of damping are applied at the top. These two damping methods are discussed in the manuscript now.

24) Line 400: An analysis is discussed, but no figure or scores are provided. Is this intentional or an oversight? If no figure was planned state '(not shown)'.
"not shown" is stated.
25) Line 409 typo 'zerio'
corrected
26) Lines 507 and 509: The phrases 'NOAA-EMC/global-workflow at gfs.v16.2.2 (github.com)' and 'Index of /data/nccf/com/gfs/prod (noaa.gov)' look like links, but they are not functional. Correct this.
The links are fixed
27) Line 541: Point to the newer version of the Harris et al FV3 model description (Technical document from June 2021). The provided link leads to an empty page. It is incorrect to also state that this was published in J. Adv. Model Earth Sys.,12 (10), 2020b (remove the journal name)
Corrected.
28) Lines 525 & 584: incomplete references, both needs the location and conference dates. Capitalize fv3gfs and fv3.
Locations and conference dates are added.
29) Figs. 4, 5, 6, 7 are rather fuzzy and of low quality. Improve the quality.

Figs 4, 5 6 7 are replotted.

---

## Author Response (AR2)

Dear Simon:

We appreciate the time you took to examine our manuscript. After conducting a careful review, we have rectified some inconsistencies and formatting errors, among other issues. Additionally, we have cited our Zenodo repository in the manuscript. We have uploaded two versions of the revised manuscript: a clean version and a revision-tracked version.

Thank you for your editorial efforts.

Best regards

Xiaqiong and Henry